# Morphological and Multi-Gene Phylogenetic Analyses Reveal *Pseudotubeufia* gen. nov. and Two New Species in Tubeufiaceae from China

**DOI:** 10.3390/jof9070742

**Published:** 2023-07-12

**Authors:** Jian Ma, Xing-Juan Xiao, Ning-Guo Liu, Saranyaphat Boonmee, Yuan-Pin Xiao, Yong-Zhong Lu

**Affiliations:** 1School of Food and Pharmaceutical Engineering, Guizhou Institute of Technology, Guiyang 550003, China; yanmajian@163.com (J.M.); juanj0826@163.com (X.-J.X.); liuningguo11@gmail.com (N.-G.L.); emmaypx@gmail.com (Y.-P.X.); 2Center of Excellence in Fungal Research, Mae Fah Luang University, Chiang Rai 57100, Thailand; saranyaphat.boo@mfu.ac.th; 3School of Science, Mae Fah Luang University, Chiang Rai 57100, Thailand

**Keywords:** three new taxa, asexual morph, taxonomy, Tubeufiales

## Abstract

Three helicosporous hyphomycete collections representing two species were obtained from rotting wood found in freshwater and terrestrial habitats in the Guizhou and Guangxi Provinces, China. A new genus *Pseudotubeufia* (Tubeufiaceae, Tubeufiales), comprising *Ps*. *hyalospora* sp. nov. and *Ps*. *laxispora* sp. nov., was introduced with morphological characteristic and molecular data. In addition, the molecular evidence showed that *Helicomyces* sp. (G.M. 2020-09-19.1), *H*. *roseus* (CBS: 102.76), and the new genus *Pseudotubeufia* clustered together with high support based on a multi-gene (LSU, ITS, *tef1α*, and *rpb2*) phylogenetic analysis. Detailed descriptions, illustrations, and notes of the three new collections are provided.

## 1. Introduction

*Tubeufia* was first introduced by Penzig and Saccardo [1], which included the type species *T*. *javanica* and two other species (*T*. *anceps* and *T*. *coronata*). Based on *Tubeufia*, the family Tubeufiaceae and order Tubeufiales were subsequently established [2,3]. The latest comprehensive study on Tubeufiaceae was carried out by Lu et al. [4]. Currently, there are 46 accepted genera in the family Tubeufiaceae, including *Acanthohelicospora*, *Acanthophiobolus*, *Acanthostigma*, *Acanthostigmina*, *Acanthotubeufia*, *Aquaphila*, *Artocarpomyces*, *Berkleasmium*, *Bifrontia*, *Boerlagiomyces*, *Chaetosphaerulina*, *Chlamydotubeufia*, *Dematiohelicoma*, *Dematiohelicomyces*, *Dematiohelicosporum*, *Dematiotubeufia*, *Dictyospora*, *Helicangiospora*, *Helicoarctatus*, *Helicodochium*, *Helicohyalinum*, *Helicoma*, *Helicomyces*, *Helicosporium*, *Helicotruncatum*, *Helicotubeufia*, *Kamalomyces*, *Kevinhydea*, *Manoharachariella*, *Muripulchra*, *Neoacanthostigma*, *Neochlamydotubeufia*, *Neohelicoma*, *Neohelicomyces*, *Neohelicosporium*, *Neomanoharachariella*, *Neotubeufia*, *Parahelicomyces*, *Pleurohelicosporium*, *Podonectria*, *Pseudohelicoon*, *Tamhinispora*, *Thaxteriella*, *Thaxteriellopsis*, *Tubeufia*, and *Zaanenomyces* [2,3,4,5,6,7,8,9,10,11,12,13,14,15,16]. Among them, five genera, *viz*. *Acanthophiobolus*, *Bifrontia*, *Boerlagiomyces*, *Podonectria*, and *Thaxteriella*, only have morphological data available, and their systematic evolutionary relationships have not been confirmed by molecular data.

Helicosporous hyphomycetes are asexual fungi that produce various forms of coiled two- or three-dimensional hollow conidia, which is the most common asexual morph in the family Tubeufiaceae [4,17,18,19,20,21]. The classification of helicosporous hyphomycetes has been studied for more than 200 years [22,23,24]. These fungi are widely distributed in tropical and subtropical regions, mostly acting as saprobes on plant litter, rotten wood, and decaying twigs in freshwater and terrestrial habitats [4,20,21]. However, there have been rare reports of endophytic fungi with coiled conidia [25,26].

In this study, three new collections from the family Tubeufiaceae were obtained during a survey of helicosporous hyphomycetes from the Guizhou and Guangxi Provinces, China. Based on detailed morphological comparisons and multi-gene phylogenetic analyses, we introduced a new genus named *Pseudotubeufia*, which comprises two new species, *Ps*. *hyalospora* and *Ps*. *laxispora*.

## 2. Materials and Methods

### 2.1. Sample Collection, Specimen Examination, and Isolation

Fresh specimens of submerged rotting wood were collected from May to August 2021 in the Guizhou and Guangxi provinces in southern China. The newly collected samples were processed following the method described by Boonmee et al. [3]. The colonies on the host surfaces were examined and observed with stereomicroscopes (SMZ 745 and SMZ 800N, Nikon, Tokyo, Japan). Their micro-morphological characters were studied using an ECLIPSE Ni compound microscope (Nikon, Tokyo, Japan) and a Canon 90D digital camera. Measurements were made with the Tarosoft (R) Image Frame Work program. Photo-plates were made using Adobe Illustrator CC 2019 (Adobe Systems, San Jose, CA, USA).

Single spores were isolated on potato dextrose agar (PDA) medium and the germinated conidia were aseptically transferred to fresh PDA plates, as described in Senanayake et al. [27]. Fungal colonies growing on the PDA were incubated at 25 °C for 28 or 42 days, and their morphological characteristics, including color and size, were recorded. Dried fungal specimens were deposited in the herbarium of the Kunming Institute of Botany, Chinese Academy of Sciences (Herb. HKAS), Kunming, China, and in the herbarium of the Guizhou Academy of Agriculture Sciences (Herb. GZAAS), Guiyang, China. Ex-type living cultures were deposited at the China General Microbiological Culture Collection Center (CGMCC), Beijing, China, and the Guizhou Culture Collection, China (GZCC). Facesoffungi numbers (FoF) and Index Fungorum numbers were determined according to the guidelines of Jayasiri et al. [28] and the Index Fungorum (2023) [29], respectively.

### 2.2. DNA Extraction, PCR Amplification, and Sequencing

Fresh fungal mycelia were scraped using the methods described by Lu et al. [30]. Genomic DNA was extracted using the Biospin Fungus Genomic DNA Extraction Kit (BioFlux, Shanghai, China), according to the manufacturer’s protocol. The large subunit of the ribosomal DNA (LSU), the internal transcribed spacer (ITS), the translation elongation factor 1 alpha (*tef1α*), and the RNA polymerase II second largest subunit (*rpb2*) gene regions were amplified using LR0R/LR5, ITS5/ITS4, EF1-983F/EF1-2218R, and fRPB2-5F/fRPB2-7cR primer pairs, respectively [31,32,33,34]. PCR amplification was performed in a reaction volume of 50 μL, including 44 μL 1.1 × T3 Supper PCR Mix (Qingke Biotech, Chongqing, China), 2 μL of each forward and reverse primer, and a 2 μL DNA template. The LSU, ITS, *tef1α*, and *rpb2* amplification reactions were carried out according to the following reference method (Table 1) [4,13,30,35,36,37].

The quality of the PCR products was checked on 1% agarose gel electrophoresis stained with ethidium bromide. The purification and sequencing of the PCR products were carried out at Tsingke Bio-logical Engineering Technology and Services Co., Ltd. (Chongqing, China).

### 2.3. Phylogenetic Analyses

The original sequences of our newly obtained strains were checked and assembled using BioEdit v 7.0.5.3 and SeqMan v. 7.0.0 (DNASTAR, Madison, WI) [38,39]. The closest taxa to our strains were determined by blast searches in GenBank (http://www.ncbi.nlm.nih.gov/, accessed on 10 May 2023). The other sequences used in the phylogenetic analysis (Table 2) were downloaded from GenBank (http://www.ncbi.nlm.nih.gov/, accessed on 10 May 2023). The sequence alignments for each locus were performed using the online multiple alignment program MAFFT version 7 (https://mafft.cbrc.jp/alignment/server/, accessed on 10 May 2023) [40], and auto-adjusted by trimAl v1.2 [41]. The multigenic sequences (LSU-ITS-*tef1α*-*rpb2*) were merged using the SequenceMatrix-Windows-1.7.8 software, and the sequences were exported to CIPRES for RAxML analyses [42]. The aligned Fasta and Phylip format file was converted to a Nexus format file for Bayesian inference (BI) and Maximum Parsimony (MP) analyses by using AliView v. 1.27 [43]. A phylogenetic tree, which infers phylogenetic relationships, was reconstructed based on a concatenated LSU, ITS, *tef1α*, and *rpb2* dataset using the online CIPRES Science Gateway (https://www.phylo.org/portal2/home.action, accessed on 10 May 2023) to construct the Maximum Likelihood (ML), Maximum Parsimony (MP), and Bayesian inference (BI), respectively.

The maximum likelihood (ML) analysis was carried out with the RAxML-HPC2 tool on XSEDE (8.2.12) using a GTRGAMMA approximation with a rapid bootstrap analysis, followed by 1000 bootstrap replicates [44].

The maximum parsimony (MP) analysis was performed by using PAUP on the XSEDE (4.a168) tool. A heuristic search with 1000 random taxa was added to infer MP trees. The value of the MaxTrees, which collapsed branches of zero length and saved all the multiple parsimonious trees, was set to 5000. The parsimony score values of the tree length (TL), consistency index (CI), retention index (RI), and homoplasy index (HI) were calculated for the trees generated under different optimum criteria. The clade stability was estimated using a bootstrap analysis with 1000 replicates, and the taxa were added for a random stepwise of each with 10 replicates [45].

The Bayesian inference (BI) analysis was conducted in MrBayes on XSEDE (3.2.7a) [46]. The best-fit substitution model GRT + I + G was determined for the LSU, ITS, *tef1α*, and *rpb2* matrix using MrModeltest 2.3 [47] under the Akaike Information Criterion (AIC). Four simultaneous Markov chains were run for 10,000,000 generations, and trees were sampled every 1000th generation. The burn-in phase was set at 25% and the remaining trees were used to calculate the posterior probabilities (PP).

The phylogenetic tree and photo-plates were created using FigTree v. 1.4.4., Adobe Illustrator CC 2019 v. 23.1.0 (Adobe Systems, San Jose, CA, USA), and Adobe PhotoShop CC 2019 (Adobe Systems, San Jose, CA, USA).

## 3. Results

### 3.1. Phylogenetic Analysis

The partial LSU-ITS-*tef1α*-*rpb2* nucleotide sequences were used to determine the phylogenetic positions of the newly obtained isolates. These sequences were concatenated to generate a sequence matrix consisting of LSU (1–843 bp), ITS (844–1548 bp), *tef1α* (1549–2460 bp), and *rpb2* (2461–3505 bp) regions. The resulting matrix comprised a total of 3505 characters for 105 taxa and two outgroups, *Botryosphaeria agaves* (MFLUCC 10–0051) and *B*. *dothidea* (CBS 115476). The total characters analyzed in the concatenated dataset were 3505, out of which, 2002 characters were constant, 273 variable characters were parsimony-uninformative, and 1230 characters were parsimony-informative. The ML, MP, and BI analyses of the concatenated LSU-ITS-*tef1α*-*rpb2* dataset yielded similar tree topologies, and the ML tree is shown in Figure 1.

In the phylogenetic analyses (Figure 1), the newly isolated strains GZCC 22–2011 and GZCC 22–2012 clustered together (95% ML/100% MP/1 PP) without a significant branch length, indicating that they are phylogenetically the same species, as *Pseudotubeufia laxispora* sp. nov*. Pseudotubeufia hyalospora* sp. nov. formed a sister clade with *Ps. laxispora* with 91% ML/100% MP/0.97 PP supports. In addition, the three strains of *Pseudotubeufia* clustered with *Helicomyces* sp. (G.M. 2020-09-19.1), *Helicomyces roseus* (CBS 102.76), and *Dematiohelicoma pulchrum* (MUCL 39827) with weak support.

### 3.2. Taxonomy

*Pseudotubeufia* J. Ma & Y.Z. Lu, gen. nov.

Index Fungorum number: IF900553; Facesoffungi number: FoF 03700.

Etymology: “*Pseudotubeufia*”, referring to the genus morphologically similar to the helicosporous asexual morph of *Tubeufia*.

*Saprobic* on the decaying wood in a freshwater stream. The sexual morph was undetermined. The asexual morph was helicosporous hyphomycetes. The *colonies* on the substratum were superficial, effuse, gregarious, and white. The *mycelium* were partly immersed, composed of hyaline to pale brown, septate, branched, and smooth hyphae. The *conidiophores* were macronematous, mononematous, erect or procumbent, flexuous, cylindrical, branched or unbranched, septate, hyaline to brown, and smooth-walled. The *conidiogenous* cells were holoblastic, mono- to polyblastic, integrated, sympodial, repeatedly geniculate, intercalary or terminal, irregularly cylindrical, denticulate, hyaline to pale brown, and smooth-walled. The *conidia* were solitary, acropleurogenous, helicoid, rounded at the tip, coiled 2–3 times, became loose in water, indistinctly septate, guttulate, hyaline, and smooth-walled.

Type species: *Pseudotubeufia hyalospora* J. Ma & Y.Z. Lu.

Notes: Morphologically, *Pseudotubeufia* is the most similar to *Tubeufia* as it has flexuous, cylindrical conidiophores, cylindrical, denticulate, hyaline to pale brown conidiogenous cells, and hyaline helicoid conidia [4]. However, the phylogenetic analysis result showed that *Pseudotubeufia* has a close affinity with the species of *Dematiohelicoma* and *Helicomyces*, and is distant from the group of *Tubeufia* (Figure 1). However, *Dematiohelicoma* can be distinguished from *Pseudotubeufia* by its erect conidiophores and multi-septate, brown to dark brown conidia. *Pseudotubeufia* is also easily distinguished from *Helicomyces* by its repeatedly geniculate conidiogenous cells [4]. Therefore, the new genus *Pseudotubeufia* is introduced to accommodate two species, *Ps*. *hyalospora* and *Ps*. *laxispora*.

*Pseudotubeufia hyalospora* J. Ma & Y.Z. Lu., sp. nov., Figure 2.

Index Fungorum number: IF900554; Facesoffungi number: FoF 14268.

Etymology: The epithet “*hyalospora*”, referring to hyaline helicoid conidia.

Holotype: HKAS 125885.

*Saprobic* on the decaying wood in a freshwater stream. The sexual morph was undetermined. The asexual morph was helicosporous hyphomycetes. The *colonies* on the substratum were superficial, effuse, gregarious, and white. The *mycelium* were partly immersed, composed of hyaline to pale brown, septate, branched, and smooth hyphae. The *conidiophores* were 31–46 μm long, 3–5.5 μm wide, macronematous, mononematous, procumbent, flexuous, cylindrical, branched, septate, hyaline to pale brown, and smooth-walled. The *conidiogenous* cells were 5.5–27.5 μm long, 3–5 μm wide, holoblastic, mono- to polyblastic, integrated, sympodial, repeatedly geniculate, intercalary or terminal, irregularly cylindrical, denticulate, hyaline to pale brown, and smooth-walled. The *conidia* were solitary, acropleurogenous, helicoid, rounded at the tip, 35–58 μm in diam. and had conidial filaments 4–5.5 μm wide ($\bar{x}$ = 48 × 4.5 μm, n = 20), 201–316 μm long, coiled 2–3 times, became loose in water, were indistinctly septate, guttulate, and hyaline.

Culture characteristics: The conidia germinated on the PDA within 10 h. The colonies on the PDA were irregular, with a flat surface, edge undulate, were pale brown to brown from above and below, and reached a 28 mm diam. after 42 days of incubation at 25 °C.

Material examined: China, Guizhou Province, Qiandongnan Miao and Dong Autonomous Prefecture, Zhenyuan City, 27°18′ N, 108°21′ E, on rotting wood in a freshwater stream, 1 May 2021, Xing-Juan Xiao, XXJ11.2 (HKAS 125885, holotype; GZAAS 22–2010, isotype), ex-type living cultures, CGMCC, GZCC 22–2010.

Notes: Morphologically, *Ps*. *hyalospora* is similar to *Ps*. *laxispora* (HKAS 125868), as it has flexuous, branched conidiophores, repeatedly geniculate conidiogenous cells, and acropleurogenous, guttulate, hyaline helicoid conidia. However, *Pseudotubeufia hyalospora* differs from *Ps*. *laxispora* (HKAS 125868) in having shorter conidiophores (31–46 μm vs. up to 155 μm), shorter conidiogenous cells (5.5–27.5 μm vs. up to 39 μm), and a different colony morphology in PDA (irregular, undulate edge vs. circular, entire edge). In addition, the phylogenetic analysis result showed that they are a distinct species. In accordance with the recommendations of Jeewon and Hyde [48] for species delimitation, we analyzed the pairwise dissimilarities of the DNA sequences between *Ps. hyalospora* (GZCC 22–2010) and *Ps*. *laxispora* (GZCC 22–2011) and found 60/905 bp (6.6%) differences in the *tef1α* gene. Therefore, we propose *Pseudotubeufia hyalospora* as a new species.

*Pseudotubeufia laxispora* J. Ma & Y.Z. Lu, sp. nov., Figure 3.

Index Fungorum number: IF900555; Facesoffungi number: FoF 14269.

Etymology: The epithet “*laxispora*”, referring to loosely coiled conidia.

Holotype: HKAS 125868.

**Holotype**: *Saprobic* on dead bamboo culms in a freshwater stream. The sexual morph was undetermined. The asexual morph was helicosporous hyphomycetes. The *colonies* on the substratum were superficial, effuse, gregarious, and white. The *mycelium* were partly immersed, composed of hyaline to pale brown, septate, and abundantly branched hyphae. The *conidiophores* were 30–155 μm long, 3.5–6.5 μm wide, macronematous, mononematous, procumbent, flexuous, irregular cylindrical, branched, septate, hyaline to pale brown, and smooth-walled. The *conidiogenous* cells were 10–39 μm long, 3.5–6 μm wide, holoblastic, mono- to polyblastic, integrated, sympodial, intercalary or terminal, cylindrical, repeatedly geniculate, hyaline to pale brown, and smooth-walled. The *conidia* were solitary, acropleurogenous, helicoid, rounded at the tip, 35–56 μm in diam. and had conidial filaments that were 3–6.5 μm wide ($\bar{x}$ = 45 × 4.5 μm, n = 20), 242–327 μm long, loosely coiled 2^1^/_4_–2^3^/_4_ times, became loosely coiled in water, were indistinctly multi-septate, guttulate, hyaline, and smooth-walled; **Paratype** (Figure 4): *Saprobic* on the decaying wood in a terrestrial habitat. The sexual morph was undetermined. The asexual morph was helicosporous hyphomycetes. The *colonies* on the substratum were superficial, effuse, gregarious, and white. The *mycelium* were partly immersed, composed of hyaline to pale brown, septate, and abundantly branched hyphae. The *conidiophores* were 21–184 μm long, 3.5–9 μm wide, macronematous, mononematous, erect, flexuous, cylindrical, branched, septate, with the lower part dark brown and the upper part hyaline to pale brown, and smooth-walled. The *conidiogenous* cells were 4.5–33.5 μm long, 3–5.5 μm wide, holoblastic, mono- to polyblastic, integrated, sympodial, with arising tiny bladder-like protrusions, intercalary or terminal, cylindrical, truncate at apex after conidial secession, hyaline to pale brown, and smooth-walled. The *conidia* were solitary, acropleurogenous, helicoid, rounded at the tip, 36–50.5 μm in diam. and had conidial filaments that were 3.5–6 μm wide ($\bar{x}$ = 42 × 4.5 μm, n = 20), 189–231 μm long, coiled 1^1^/_2_–2^1^/_2_ times, became loosely coiled in water, were indistinctly multi-septate, guttulate, hyaline, and smooth-walled.

Culture characteristics: Holotype: The conidia germinated on the PDA within 10 h. The colonies on the PDA were circular, with a flat surface, edge entire, pale brown to brown from above and below, and reached 33 mm in diam. after 42 days of incubation at 25 °C; Paratype: The conidia germinated on the PDA within 10 h. The colonies on the PDA were circular, with a flat surface, edge entire, dark brown from above and below, and reached 22 mm in diam. after 28 days of incubation at 25 °C.

Material examined: China, Guangxi Province, Liuzhou City, Luzhai County, 24°46′ N, 109°53′ E, on dead bamboo culms in a freshwater stream, 4 May 2021, Jian Ma, LZ6.2 (HKAS 125868, holotype; GZAAS 22–2011, isotype), ex-type living cultures, CGMCC, GZCC 22–2011; China, Guizhou Province, Qiannan Buyi and Miao Autonomous Prefecture, Sandu City, 25°56′ N, 107°57′ E, on decaying wood in a terrestrial habitat, 12 August 2021, Jingyi Zhang, SD12 (GZAAS 22–2012; paratype), living culture GZCC 22–2012.

Notes: Two collections, HKAS 125868 and GZAAS 22–2012, were obtained from freshwater and terrestrial habitats in southern China. Morphologically, HKAS 125868 has procumbent and hyaline conidiophores, while GZAAS 22–2012 has erect and brown conidiophores. Additionally, GZAAS 22–2012 has smaller conidia compared to HKAS 125868 (189–231 μm vs. 242–327 μm). However, based on pairwise nucleotide comparisons of ITS, LSU, *tef1α*, and *rpb2*, GZCC 22–2011 only differs from GZCC 22–2012 in a few genetic markers (2/469 bp for ITS, 1/824 bp for LSU, 1/916 bp for *tef1α*, and 13/1113 bp for *rpb2*). Furthermore, the phylogenetic analysis did not show any significant differences between these two strains (Figure 1). Therefore, despite their distinct morphology, we introduce these two isolates as one species named *Pseudotubeufia laxispora*.

## 4. Conclusions

In this study, we introduced a new genus, *Pseudotubeufia*, based on multi-gene phylogenetic analyses and morphological characteristics. Morphologically, the asexual morphs of *Ps. hyalospora* and *Ps*. *laxispora* (HKAS 125868) are most similar to the species of *Tubeufia*, while *Ps*. *laxispora* (GZAAS 22–2012) resembles the species of *Parahelicomyces*. However, the multi-gene phylogenetic analyses showed that they did not cluster with *Tubeufia* or *Parahelicomyces*. Instead, they formed a distinct sister clade with the strains *Helicomyces* sp. (G.M. 2020-09-19.1, GenBank: MW276143) and *H*. *roseus* (CBS: 102.76), with 100% ML/100% MP/1 PP supports (Figure 1).

The ITS sequences of *Ps*. *hyalospora* and *Ps*. *laxispora* were searched using BLASTn in NCBI GenBank, and they exhibited the highest similarities to *Helicomyces* sp. (G.M. 2020-09-19.1: 58% query cover, 97.49% similarity and 100% query cover, 97.53% similarity), *Helicomyces roseus* (CBS 102.76: 58% query cover, 97.11% similarity and 100% query cover, 97.35% similarity), and *Tubeufia* sp. (MFLUCC 17–1520 and KUMCC 21–0472: 97% query cover, 84.98% similarity and 99% query cover, 86% similarity), respectively. In order to confirm the phylogenetic positions of the newly isolated strains, we performed single-gene and multi-gene phylogenetic analyses, including all species of the genera *Tubeufia*, *Parahelicomyces*, *Helicomyces*, and other related taxa, and obtained the same conclusion as shown in Figure 1. It is worth noting that *Helicomyces* sp. (G.M. 2020-09-19.1) and *H*. *roseus* (CBS 102.76) currently lack morphological descriptions and only have molecular data [49]. Their taxonomic positions require further molecular data and morphological descriptions for clarification.

Morphological differences can vary widely, even within the same species of helicosporous hyphomycetes. For instance, two collections (MFLU 16–2544 from decaying wood in China and MFLU 17–1091 from decaying wood in Thailand) have been identified as the same species, namely *Tubeufia aquatica* [4,50]. However, MFLU 16–2544 has larger conidiophores (109.5–189.5 μm) than those of MFLU 17–1091 (18–40 μm). Additionally, the conidiophores of MFLU 16–2544 are multi-septate, branched, and brown to dark brown, while those of MFLU 17–1091 are 0–1-septate, unbranched, and pale brown [4,50]. Similarly, our two collections of *Ps*. *laxispora* (HKAS 125868 and GZAAS 22–2012) showed significant differences in their conidiophores (Figure 3 and Figure 4). We speculate that such differences may be attributable to variations in their habitats and geographical regions.

## Figures and Tables

**Figure 1 jof-09-00742-f001:**
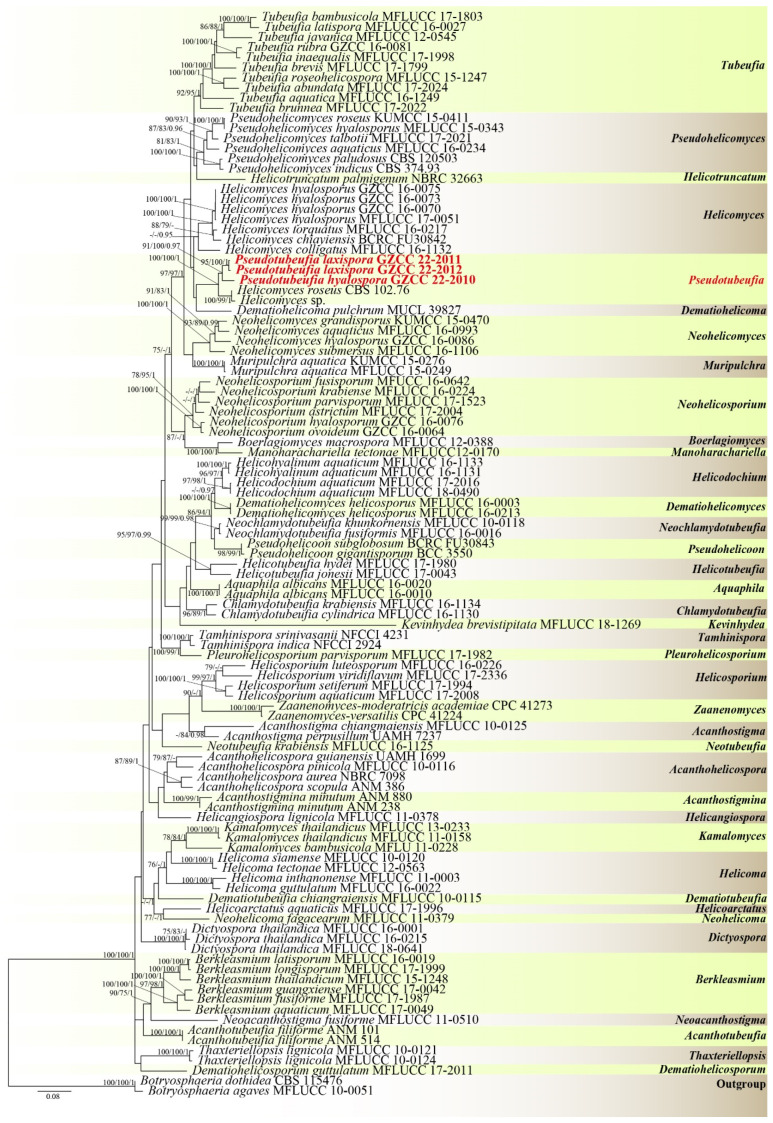
Phylogram generated from the maximum likelihood analysis based on a combined of LSU, ITS, *tef1α*, and *rpb2* sequence data. Bootstrap support values of maximum likelihood (ML) and maximum parsimony (MP) equal to or greater than 75%, and Bayesian posterior probabilities (PP) equal to or greater than 0.95 are given near the nodes as ML/MP/PP. *Botryosphaeria agaves* (MFLUCC 10–0051) and *B*. *dothidea* (CBS 115476) were used as outgroup taxa. The newly generated sequences are shown in red bold.

**Figure 2 jof-09-00742-f002:**
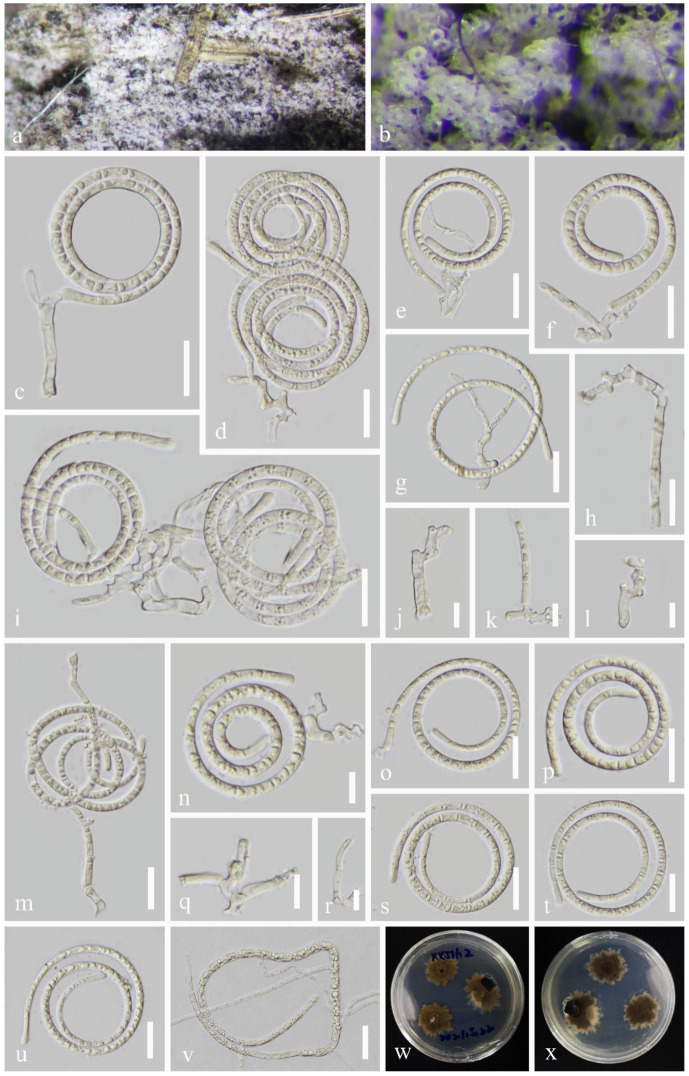
*Pseudotubeufia hyalospora* (HKAS 125885, holotype). (**a**,**b**) Colonies on the host surface. (**c**–**g**,**i**,**m**,**n**) Conidiophores with attached conidia. (**h**,**j**–**l**,**q**,**r**) Conidiophores and conidiogenous cells. (**o**,**p**,**s**–**u**) Conidia. (**v**) Germinating conidium. (**w**,**x**) Colonies on PDA at 42 days old (from above and below). Scale bars: (**c**–**i**,**m**,**o**,**p**,**s**–**v**) 20 µm, (**j**–**l**,**n**,**q**,**r**) 10 µm.

**Figure 3 jof-09-00742-f003:**
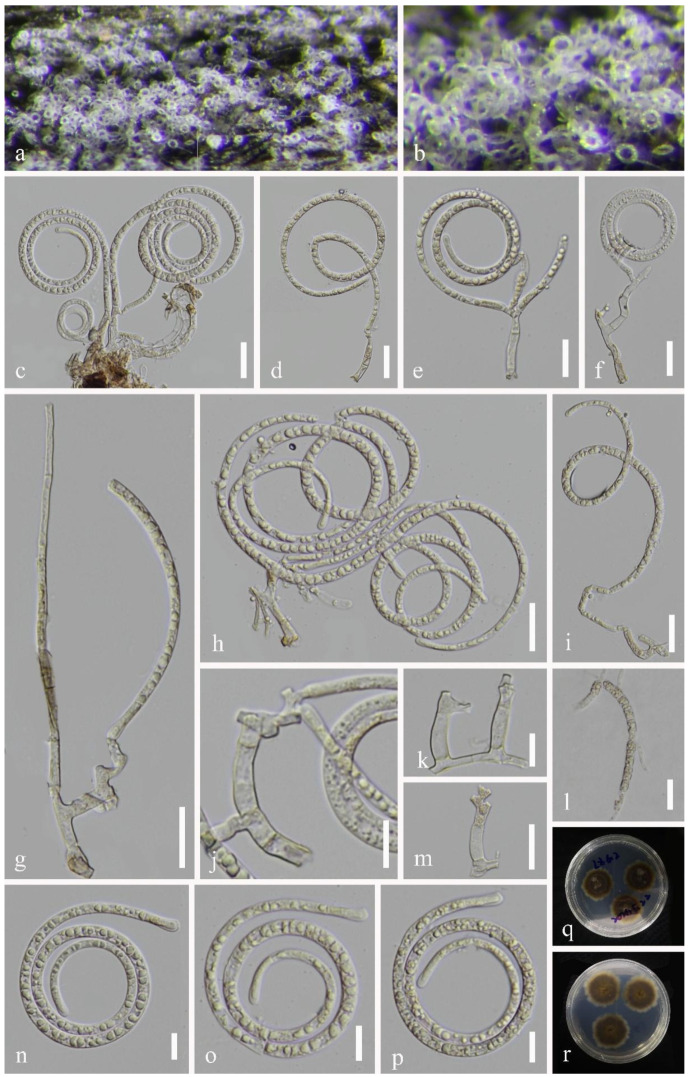
*Pseudotubeufia laxispora* (HKAS 125868, holotype). (**a**,**b**) Colonies on the host surface. (**c**–**i**) Conidiophores with attached conidia. (**j**,**k**,**m**) Conidiogenous cells. (**l**) Germinating conidium. (**n**–**p**) Conidia. (**q**,**r**) Colonies on PDA at 42 days old (from above and below). Scale bars: (**c**–**i**,**l**,**m**) 20 µm, (**j**,**k**,**n**–**p**) 10 µm.

**Figure 4 jof-09-00742-f004:**
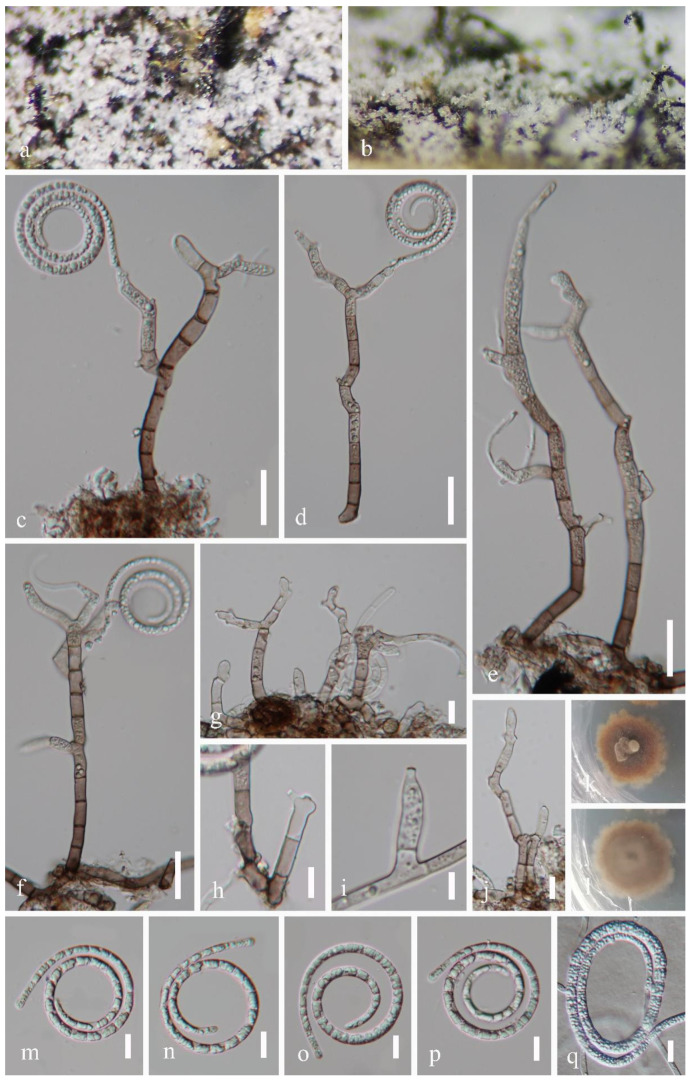
*Pseudotubeufia laxispora* (GZAAS 22–2012, paratype). (**a**,**b**) Colonies on host surface. (**c**–**g**,**j**) Conidiophores with attached conidia. (**h**,**i**) Conidiogenous cells. (**m**–**p**) Conidia. (**q**) Germinating conidium. (**k**,**l**) Colonies on PDA at 28 days old (from above and below). Scale bars: (**c**–**f**) 20 µm, (**g**,**h**,**j**–**q**) 10 µm, (**i**) 5 µm.

**Table 1 jof-09-00742-t001:** PCR protocols.

Locus	Primer	Initial Denaturation	Denaturation	Annealing	Elongation	Final Extension	Hold
LSU	LR0R/LR5	95 °C/3 min	94 °C/30 s	51 °C/50 s	72 °C/1 min	72 °C/7 min	4 °C
			30 cycles				
ITS	ITS5/ITS4	95 °C/3 min	95 °C/30 s	51 °C/1 min	72 °C/45 s	72 °C/10 min	
			34 cycles				
*tef1α*	EF1-983F/EF1-2218R	95 °C/3 min	94 °C/30 s	55 °C/50 s	72 °C/1 min	72 °C/7 min	
			40 cycles				
*rpb2*	fRPB2-5F/fRPB2-7cR	95 °C/3 min	95 °C/30 s	54 °C/40 s	72 °C/1 min	72 °C/7 min	
			34 cycles				

**Table 2 jof-09-00742-t002:** Taxa used in this study and their GenBank accession numbers.

Taxon	Strain	GenBank Accessions			
		LSU	ITS	*tef1α*	*rpb2*
*Acanthohelicospora aurea*	NBRC 7098	AY856894	AY916478	–	–
*Acanthohelicospora guianensis*	UAMH 1699	AY856891	AY916479	–	–
*Acanthohelicospora pinicola*	MFLUCC 10-0116 ^T^	KF301534	KF301526	KF301555	–
*Acanthohelicospora scopula*	ANM 386	GQ850489	GQ856141	–	–
*Acanthostigma chiangmaiensis*	MFLUCC 10-0125 ^T^	JN865197	JN865209	KF301560	–
*Acanthostigma perpusillum*	UAMH 7237	AY856892	AY916492	–	–
*Acanthostigmina minutum*	ANM 238	GQ850487	–	–	–
*Acanthostigmina minutum*	ANM 880	GQ850486	–	–	–
*Acanthotubeufia filiforme*	ANM 101 ^T^	GQ850495	–	–	–
*Acanthotubeufia filiforme*	ANM 514	GQ850494	GQ856146	–	–
*Aquaphila albicans*	MFLUCC 16-0010	KX454166	KX454165	KY117034	MF535255
*Aquaphila albicans*	MFLUCC 16-0020	KX454168	KX454167	–	MF535256
*Berkleasmium aquaticum*	MFLUCC 17-0049 ^T^	KY790432	KY790444	KY792608	MF535268
*Berkleasmium fusiforme*	MFLUCC 17-1987 ^T^	MH558822	MH558695	MH550886	MH551009
*Berkleasmium guangxiense*	MFLUCC 17-0042 ^T^	KY790436	KY790448	KY792612	MF535270
*Berkleasmium latisporum*	MFLUCC 16-0019 ^T^	KY790437	KY790449	KY792613	MF535271
*Berkleasmium longisporum*	MFLUCC 17-1999 ^T^	MH558825	MH558698	MH550889	MH551012
*Berkleasmium thailandicum*	MFLUCC 15-1248 ^T^	MH558829	KX454176	KY792614	MH551017
*Boerlagiomyces macrospora*	MFLUCC 12-0388	KU764712	KU144927	KU872750	–
*Botryosphaeria agaves*	MFLUCC 10-0051	JX646807	JX646790	–	–
*Botryosphaeria dothidea*	CBS 115476	DQ678051	KF766151	DQ767637	DQ677944
*Chlamydotubeufia cylindrica*	MFLUCC 16-1130 ^T^	MH558830	MH558702	MH550893	MH551018
*Chlamydotubeufia krabiensis*	MFLUCC 16-1134	KY678759	KY678767	KY792598	MF535261
*Dematiohelicoma pulchrum*	MUCL 39827	AY856872	AY916457	–	–
*Dematiohelicomyces helicosporus*	MFLUCC 16-0213 ^T^	KX454170	KX454169	KY117035	MF535258
*Dematiohelicomyces helicosporus*	MFLUCC 16-0003	MH558831	MH558703	MH550894	MH551019
*Dematiohelicosporum guttulatum*	MFLUCC 17-2011 ^T^	MH558833	MH558705	MH550896	MH551021
*Dematiotubeufia chiangraiensis*	MFLUCC 10-0115 ^T^	JN865188	JN865200	KF301551	–
*Dictyospora thailandica*	MFLUCC 16-0001 ^T^	KY873622	KY873627	KY873286	MH551023
*Dictyospora thailandica*	MFLUCC 18-0641	MH558834	MH558706	MH550897	MH551022
*Dictyospora thailandica*	MFLUCC 16-0215	KY873623	KY873628	KY873287	–
*Helicangiospora lignicola*	MFLUCC 11-0378 ^T^	KF301531	KF301523	KF301552	–
*Helicoarctatus aquaticus*	MFLUCC 17-1996 ^T^	MH558835	MH558707	MH550898	MH551024
*Helicodochium aquaticum*	MFLUCC 17-2016 ^T^	MH558837	MH558709	MH550900	MH551026
*Helicodochium aquaticum*	MFLUCC 18-0490	MH558838	MH558710	MH550901	MH551027
*Helicohyalinum aquaticum*	MFLUCC 16-1131 ^T^	KY873620	KY873625	KY873284	MF535257
*Helicohyalinum aquaticum*	MFLUCC 16-1133 ^T^	MH558840	MH558712	MH550903	MH551029
*Helicoma guttulatum*	MFLUCC 16-0022 ^T^	KX454172	KX454171	MF535254	MH551032
*Helicoma inthanonense*	MFLUCC 11-0003 ^T^	JN865199	JN865211	–	–
*Helicoma siamense*	MFLUCC 10-0120 ^T^	JN865192	JN865204	KF301558	–
*Helicoma tectonae*	MFLUCC 12-0563 ^T^	KU764713	KU144928	KU872751	–
*Helicomyces* sp.	G.M. 2020-09-19.1	–	MW276143	–	–
*Helicomyces chiayiensis*	BCRC FU30842 ^T^	–	LC316604	–	–
*Helicomyces colligatus*	MFLUCC 16-1132	MH558853	MH558727	MH550918	MH551043
*Helicomyces hyalosporus*	MFLUCC 17-0051 ^T^	MH558857	MH558731	MH550922	MH551047
*Helicomyces hyalosporus*	GZCC 16-0070	MH558854	MH558728	MH550919	MH551044
*Helicomyces hyalosporus*	GZCC 16-0073	MH558855	MH558729	MH550920	MH551045
*Helicomyces hyalosporus*	GZCC 16-0075	MH558856	MH558730	MH550921	MH551046
*Helicomyces roseus*	CBS: 102.76	MH872733	MH860964	–	–
*Helicomyces torquatus*	MFLUCC 16-0217	MH558858	MH558732	MH550923	MH551048
*Helicosporium aquaticum*	MFLUCC 17-2008 ^T^	MH558859	MH558733	MH550924	MH551049
*Helicosporium luteosporum*	MFLUCC 16-0226 ^T^	KY321327	KY321324	KY792601	MH551056
*Helicosporium setiferum*	MFLUCC 17-1994 ^T^	MH558861	MH558735	MH550926	MH551051
*Helicosporium viridiflavum*	MFLUCC 17-2336 ^T^	–	MH558738	MH550929	MH551054
*Helicotruncatum palmigenum*	NBRC 32663	AY856898	AY916480	–	–
*Helicotubeufia hydei*	MFLUCC 17-1980 ^T^	MH290026	MH290021	MH290031	MH290036
*Helicotubeufia jonesii*	MFLUCC 17-0043 ^T^	MH290025	MH290020	MH290030	MH290035
*Kamalomyces bambusicola*	MFLU 11-0228 ^T^	MF506880	–	–	–
*Kamalomyces thailandicus*	MFLUCC 11-0158	MF506881	MF506883	MF506885	–
*Kamalomyces thailandicus*	MFLUCC 13-0233 ^T^	MF506882	MF506884	MF506886	–
*Kevinhydea brevistipitata*	MFLUCC 18-1269 ^T^	MH747115	MH747102	–	–
*Manoharachariella tectonae*	MFLUCC12-0170 ^T^	KU764705	KU144935	KU872762	–
*Muripulchra aquatica*	KUMCC 15-0276	KY320551	KY320534	KY320564	MH551058
*Muripulchra aquatica*	MFLUCC 15-0249 ^T^	KY320549	KY320532	–	–
*Neoacanthostigma fusiforme*	MFLUCC 11-0510 ^T^	KF301537	KF301529	–	–
*Neochlamydotubeufia fusiformis*	MFLUCC 16-0016 ^T^	MH558865	MH558740	MH550931	MH551059
*Neochlamydotubeufia khunkornensis*	MFLUCC 10-0118 ^T^	JN865190	JN865202	KF301564	–
*Neohelicoma fagacearum*	MFLUCC 11-0379 ^T^	KF301532	KF301524	KF301553	–
*Neohelicomyces aquaticus*	MFLUCC 16-0993 ^T^	KY320545	KY320528	KY320561	MH551066
*Neohelicomyces grandisporus*	KUMCC 15-0470 ^T^	KX454174	KX454173	–	MH551067
*Neohelicomyces hyalosporus*	GZCC 16-0086 ^T^	MH558870	MH558745	MH550936	MH551064
*Neohelicomyces submersus*	MFLUCC 16-1106 ^T^	KY320547	KY320530	–	MH551068
*Neohelicosporium astrictum*	MFLUCC 17-2004 ^T^	MH558872	MH558747	MH550938	MH551070
*Neohelicosporium fusisporum*	MFUCC 16-0642 ^T^	MG017613	MG017612	MG017614	–
*Neohelicosporium hyalosporum*	GZCC 16-0076 ^T^	MF467936	MF467923	MF535249	MF535279
*Neohelicosporium krabiense*	MFLUCC 16-0224 ^T^	MH558879	MH558754	MH550945	MH551077
*Neohelicosporium ovoideum*	GZCC 16-0064 ^T^	MH558881	MH558756	MH550947	MH551079
*Neohelicosporium parvisporum*	MFLUCC 17-1523 ^T^	MF467939	MF467926	MF535252	MF535282
*Neotubeufia krabiensis*	MFLUCC 16-1125 ^T^	MG012024	MG012031	MG012010	MG012017
*Parahelicomyces aquaticus*	MFLUCC 16-0234 ^T^	MH558891	MH558766	MH550958	MH551092
*Parahelicomyces hyalosporus*	MFLUCC 15-0343 ^T^	KY320540	KY320523	–	–
*Parahelicomyces indicus*	CBS 374.93	AY856885	AY916477	–	–
*Parahelicomyces paludosus*	CBS 120503	DQ341103	DQ341095	–	–
*Parahelicomyces roseus*	KUMCC 15-0411	KY320544	KY320527	KY320560	–
*Parahelicomyces talbotii*	MFLUCC 17-2021	MH558890	MH558765	MH550957	MH551091
*Pleurohelicosporium parvisporum*	MFLUCC 17-1982 ^T^	MH558889	MH558764	MH550956	MH551088
*Pseudohelicoon gigantisporum*	BCC 3550	AY856904	AY916467	–	–
*Pseudohelicoon subglobosum*	BCRC FU30843 ^T^	LC316610	LC316607	–	–
* Psedotubeufia laxispora *	GZCC 22-2011 ^T^	OR030831	OR030838	OR046675	OR046682
* Psedotubeufia laxispora *	GZCC 22-2012	OR030832	OR030839	OR046676	OR046683
* Psedotubeufia hyalospora *	GZCC 22-2010 ^T^	OR030833	OR030840	OR046677	–
*Tamhinispora indica*	NFCCI 2924 ^T^	KC469283	KC469282	–	–
*Tamhinispora srinivasanii*	NFCCI 4231 ^T^	MG763745	MG763746	–	–
*Thaxteriellopsis lignicola*	MFLUCC 10-0121	JN865193	JN865205	–	–
*Thaxteriellopsis lignicola*	MFLUCC 10-0124	JN865196	JN865208	KF301561	–
*Tubeufia abundata*	MFLUCC 17-2024 ^T^	MH558894	MH558769	MH550961	MH551095
*Tubeufia aquatica*	MFLUCC 16-1249 ^T^	KY320539	KY320522	KY320556	MH551142
*Tubeufia bambusicola*	MFLUCC 17-1803 ^T^	MH558896	MH558771	MH550963	MH551097
*Tubeufia brevis*	MFLUCC 17-1799 ^T^	MH558897	MH558772	MH550964	MH551098
*Tubeufia brunnea*	MFLUCC 17-2022 ^T^	MH558898	MH558773	MH550965	MH551099
*Tubeufia inaequalis*	MFLUCC 17-1998 ^T^	MH558916	MH558791	MH550984	MH551117
*Tubeufia javanica*	MFLUCC 12-0545 ^T^	KJ880036	KJ880034	KJ880037	–
*Tubeufia latispora*	MFLUCC 16-0027 ^T^	KY092412	KY092417	KY117033	MH551119
*Tubeufia roseohelicospora*	MFLUCC 15-1247 ^T^	KX454178	KX454177	–	MH551144
*Tubeufia rubra*	GZCC 16-0081 ^T^	MH558926	MH558801	MH550994	MH551128
*Zaanenomyces moderatricis-academiae*	CPC 41273 ^T^	OK663762	OK664723	–	OK651167
*Zaanenomyces versatilis*	CPC 41224 ^T^	OK663769	OK664730	–	–

Note: Newly generated sequences in this study are indicated in blue bold. “^T^” denotes ex-type strain. “–” as meaning no data available in GenBank.

## Data Availability

All sequences generated in this study were submitted to GenBank database.

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
