# Peer review of "Morphological and Multi-Gene Phylogenetic Analyses Reveal Pseudotubeufia gen. nov. and Two New Species in Tubeufiaceae from China"

_jof, 2023, doi:10.3390/jof9070742_

Round 1

Reviewer 1 Report

This manuscript introduces three new taxa of fungi in Tubeufiaceae based on morphological and molecular data. While the content merits publication, the manuscript could be strengthened by correcting some typographical errors before its publication.

Page 4                   In Table 1, change Helicomyces sp -----> Helicomyces sp.

Line no. 105         Botryosphaeria agaves should be Botryosphaeria dothidea

Line no. 113         Fig. 1 should be in full -----> Figure 1

Page 7                   Phylogenetic tree in Figure 1 needs to correct all specific epithets of Pseudotubufia

                              Pseudotubufia guttulate -----> Pseudotubufia laxispora

                              Pseudotubufia hyalosporus -----> Pseudotubufia hyalospora

Line no. 127         helicosprous -----> helicosporous

Line no. 144         Fig. 1 should be in full -----> Figure 1

Line no. 152         hyalosporus ----->  hyalospora

Line no. 167         28 cm should be 28 mm?

Line no. 220         33 cm should be 33 mm?

Line no. 222         22 cm should be 22 mm?

Line no. 238         Fig. 1 should be in full -----> Figure 1

Line no. 260         Fig. 1 should be in full -----> Figure 1

References:           some journal names are in abbreviated form, please double-checked. Scientific names need to be italicized in line no. 361, 365

Line no. 312         Medicina Tropical De So Paulo ----->  Medicina Tropical De Sao Paulo 

Author Response

Thank you very much for your reply regarding our manuscript entitled “Morphological and multi-gene phylogenetic analyses reveal Pseudotubeufia gen. nov. and two new species in Tubeufiaceae from China”. These comments are all valuable and quite helpful for revising and improving our paper. We have accepted and modified the text according to the reviewer’s comments. Revised parts are highlighted in the revision manuscript (answers to reviewer 1 are highlighted in green, answers to reviewer 2 are highlighted in red, and answers to reviewer 3 are highlighted in grey, and other changes we have made are highlighted in yellow). Attached (List of changes) is our point-by-point response to your comments.

Yours sincerely,

Jian Ma, Li-Juan Zhang, Saranyaphat Boonmee, Xing-Juan Xiao, Ning-Guo Liu, Yuan-Pin Xiao, Zong-Long Luo, Yong-Zhong Lu*

Reviewer 2 Report

The present paper describes a new genus (Pseudotubeufia gen. nov.) and new species (Ps. hyalospora sp. nov. and Ps. laxispora sp. nov) based on phylogenies of four loci (LSU, ITS, tef1α, and rpb2) and morphological characters. The paper is well written.

Minor comments are added into the manuscript.

Attached is the file with my comments and corrections.

Author Response

(The authors gave the same response as above.)

Reviewer 3 Report

The authors studied fungal strains isolated from decaying wood in Guizhou and Guangxi provinces in southern China. They introduce a new genus with two new species based on morphological and molecular data. The work presented is very good. A few minor issues can be found in the attached manuscript.

Author Response

(The authors gave the same response as above.)
